# Comparison of Radiographic Outcomes Assessed via the Radiographic Union Scale for Tibial Fractures and Alkaline Phosphatase Levels during the Tibial Healing Process: A Series of Case Reports

**DOI:** 10.3390/biology13060407

**Published:** 2024-06-03

**Authors:** André Felipe Ninomiya, Vanessa Bertolucci, Luisa Oliveira Kaneko, Nilson Nonose, Luiza di Loreto Abreu, Gabriel Rodrigues Harfuch, Ivan Gustavo Masselli dos Reis, Pedro Paulo Menezes Scariot, Leonardo Henrique Dalcheco Messias

**Affiliations:** 1Centre of Orthopedics Research, São Francisco University Hospital, Bragança Paulista 12916-900, SP, Brazil; andreninomiya@gmail.com (A.F.N.); nilsonnonose@gmail.com (N.N.); luiza_loreto@hotmail.com (L.d.L.A.); harfuchgabriel@gmail.com (G.R.H.); 2Research Group on Technology Applied to Exercise Physiology—GTAFE, Health Sciences Postgraduate Program, São Francisco University, Bragança Paulista 12916-900, SP, Brazil; vanessa.bertolucci@mail.usf.edu.br (V.B.); luisaokaneko@gmail.com (L.O.K.); ivan.reis@usf.edu.br (I.G.M.d.R.); pedroppms@yahoo.com.br (P.P.M.S.)

**Keywords:** bone consolidation, osteoblasts, X-ray

## Abstract

**Simple Summary:**

Fractures in the long bones are common injuries caused by accidents. This study looked at how these injuries heal after surgery. Two factors were checked: a substance in the blood called alkaline phosphatase and X-rays of the leg. After surgery, we found that the level of alkaline phosphatase went up, representing the bone healing. Along with this result, the X-rays also suggested the same information, that is, bone healing. We consider that by checking both the blood and the X-rays, doctors can better understand how leg bones heal after surgery. However, further studies with more patients are necessary to confirm this finding.

**Abstract:**

Background/Objectives: Tibial diaphysis fractures are common injuries resulting from high-to-low-energy traumas in patients of all age groups, but few reports currently provide complementary parameters for the assessment of bone healing processes in the postoperative period. Serum alkaline phosphatase (ALP) and the scores from the Radiographic Union Scale for Tibial Fractures (RUST) can promote new horizons in this context. Therefore, the aim of this study was to assess the behavior of ALP and RUST through within-subject comparisons from immediately post-surgery to 49 days after tibial diaphysis fracture repair. Methods: This article included four case studies where patients underwent the same procedures. Adults of both sexes aged 18 to 60 years with tibial fractures requiring surgery were included. After surgical intervention (T1), the patients were followed for 49 days after surgery, returning for follow-up appointments on the 21st (T2) and 49th (T3) days. At the follow-up appointments, new X-ray images were obtained, and blood samples were collected for ALP measurement. Results: Serum ALP levels increased by T2 following tibial reamed intramedullary nailing surgery. While this increase persisted into T3 for two patients, a decline was observed during the same period for the other two patients. Both events are indicative of the bone consolidation process, and RUST scores at the T3 corroborate this perspective for all patients included in this study. Considering that delta ALP (T3-T1 value) was lower in patients who exhibited the highest RUST score, we suggest that a synchronized analysis between ALP and RUST allows medics to diagnose bone consolidation. Conclusions: Therefore, it can be concluded that the analysis of ALP alongside RUST may be complementary for evaluating bone consolidation following tibial reamed intramedullary nailing surgery, but future studies are needed to confirm this assertion.

## 1. Introduction

Tibial diaphysis fractures are common injuries resulting from high-to-low-energy traumas in patients of all age groups [1,2,3]. These fractures are often associated with traffic accidents, with approximately 50% of them resulting from motor vehicle accidents and 24% presenting as open injuries [4,5]. In skeletally mature patients, the predominant treatment for displaced fractures of the tibial diaphysis is intramedullary nail osteosynthesis [6,7]. Despite advances in surgical techniques, local anatomical conditions may contribute to the development of complications such as delayed union and nonunion [8]. Nonunion is characterized after nine months by no signs of healing in the previous three months [9,10,11], with the incidence after internal fixation with an intramedullary nail varying between 0% and 15% [12]. Therefore, the early identification of complications in the tibia healing process is crucial for effective management and improved outcomes for patients.

Accurate radiological assessment of the stage of healing in tibial fractures has been a topic of debate [13,14,15]. The Radiographic Union Scale for Tibial Fractures (RUST) is commonly accepted to evaluate bone consolidation [8,16,17]. Based on two orthogonal radiographic views, this scale provides a final score that is associated with bone healing. Studies show that RUST is a simple, reliable, systematic, and continuous indicator in assessing tibial fractures treated with intramedullary nails [8,18]. However, the association between RUST score and the biochemical parameters commonly measured in plasma has not yet been explored. 

Alkaline phosphatase (ALP) is a membrane-bound enzyme that exhibits different structural forms depending on its location. When anchored to the cell surface, ALP typically presents as a tetrameric structure. However, once released into the systemic circulation via the action of phospholipases C and D, it assumes a dimeric form [19]. Elevated serum levels of ALP are indicative of increased osteoblastic activity, which plays a crucial role in bone matrix formation and mineralization [20,21]. Restricted to diaphyseal fracture cases, ALP was associated with the progression of the healing process [22], emerging as a clinical biomarker for the management of patients who had sustained fractures.

Although RUST scores are commonly utilized in the orthopedic clinical setting, the literature does not provide associations between these results and the serum levels of ALP. This analysis may potentially enhance both RUST and ALP utilization for bone consolidation assessment. Therefore, the aim of this study was to assess the behavior of ALP and RUST through within-subject comparisons from immediately post-surgery to 49 days after tibial diaphysis fracture repair. To this end, we present four case reports as well as initial group comparative analyses that align with the aforementioned objective, demonstrating that both ALP and RUST levels may be valid for monitoring tibial fracture recovery.

## 2. Materials and Methods

### 2.1. Case Series

This article presents four case studies where patients underwent the same procedures. Patient selection was based on the American Society of Anesthesiologists (ASA) classification, including ASA I and II and excluding ASA III to VI. Adults of both sexes, aged 18 to 60 years, with a surgical tibial fracture, were included. Pregnant women, patients with renal diseases or conditions preventing surgery, and with diagnoses other than diaphyseal fracture were excluded. Participants were informed about the study, given the freedom to withdraw, and signed a Consent Form. The study was approved by the Research Ethics Committee (registration code: 69533123.3.0000.5514) and conducted in accordance with the Declaration of Helsinki.

Although the etiologies of the fractures were distinct, all surgeries were performed at the São Francisco de Assis University Hospital. The hospital is a reference in the Bragança Paulista region, a city located 87.9 km from the capital, São Paulo. As standard procedure, the hospital conducts an initial X-ray at admission, along with a medical history interview and blood collection for hemogram analysis, which was performed using the Pentra 80 (Horiba, Irvine, CA, USA) equipment. Surgical intervention occurred within a two-day window from the patient’s admission. As part of the surgical procedure (T1), another blood sample was collected for ALP analysis (in the intraoperative period), as well as radiography at the end of the surgery for RUST evaluation. The patients were followed for 49 days after surgery, returning for follow-up appointments on the 21st (T2) and 49th (T3) days. At the follow-up appointments, new X-ray images were obtained and blood samples were collected for ALP measurement (Figure 1).

### 2.2. Tibial Reamed Intramedullary Nailing Surgery

All patients were operated on by the same surgeon (AFN), following the technique of tibial reamed intramedullary nailing. The patients were positioned in the dorsal decubitus position, with the hip and knee flexed at 90 degrees and the foot secured to the table for traction if necessary. After closed reduction of the fracture with the aid of an image intensifier, an approximately 3 cm access incision was made above the anterior tuberosity of the tibia. Using a cortical starter, an entry hole perpendicular to the tibia was made, followed by reaming of the canal. The intramedullary rod was introduced horizontally up to 2 cm from the ankle joint, with continuous radiological control. Proximal and distal locking of the rod were performed, followed by wound closure and the application of compressive dressings [23]. No complications occurred during the surgeries and the patients were subsequently transferred to the intensive care unit, where they were discharged from hospital following a thorough evaluation by the attending surgeon. 

### 2.3. X-ray

The radiographic images were acquired following the guidelines provided by Morais and Siqueira [24]. For the anteroposterior (AP) images, the patient was positioned either sitting or supine on the examination table. The leg to be radiographed was fully extended over the cassette, ensuring adequate inclusion of the knee and ankle joints, while the lower limb was positioned in Ferguson rotation. The central ray was aligned perpendicular to the cassette/chassis, precisely targeting the medial region of the leg. A 35 × 43 cm cassette/chassis was used, oriented longitudinally without the use of an anti-scatter grid, with the source-to-image receptor distance fixed at 1.00 m.

For the lateral position (LAT), the patient was positioned in lateral decubitus on the examination table, with the lateral aspect of the leg to be radiographed in direct contact with the cassette. The central ray was again directed perpendicular to the cassette/chassis, targeting the medial region of the leg. The cassette/chassis used maintained the characteristics of the previous positioning. The areas of interest, including the knee and ankle joints, as well as the tibia and fibula, were adequately visualized. This positioning procedure was specifically indicated for diagnostic and surgical control of the diaphyseal fracture of the tibia, providing an accurate radiographic representation of the injury and adjacent structures.

### 2.4. Radiographic Union Scale for Tibial Fractures

Radiograph analysis was conducted by two surgeons (AFN and NN) with over ten years of experience and specialization accredited by the Brazilian Society of Orthopedics and Traumatology [8,16,17,25]. Scores ranging from 1 to 3 for each cortical bone on orthogonal projections were attributed and totaled, resulting in a final value for each set of films. A minimum score of 4 indicated that the fracture was completely unhealed, while a maximum score of 12 indicated a fully healed fracture. A score equal to or greater than 7 indicated a minimum of three cortices with callus formation [16]. An example is shown in panel B of Figure 1.

### 2.5. Alkaline Phosphatase Measurement

After a minimum fasting period of 7 h with ad libitum water consumption, 4 mL of arterial blood was collected using tubes containing heparin for total ALP analysis. The collections were carried out under specific conditions, maintaining the pH at 10.4 and the temperature at 37 °C. Analysis of serum ALP was conducted spectrophotometrically at a wavelength of 590 nm using the LabTest kit-Ref 40-MS10009010081 (LabTest Diagnóstica, Vista Alegre, MG, Brazil) following an adaptation by Roy [26].

### 2.6. Statistical Analysis

The data from the case reports were compared relative to the T1. For group comparison, results are presented as the mean and standard deviation. Following confirmation of the normality of variance using Shapiro–Wilk test, repeated-measures ANOVAs with Bonferroni post hoc tests were employed to compare ALP levels across the three time points, in addition to assessing percentage differences and effect size (ES). The effect size criteria were defined as small (=0.2), medium (=0.5), or large (=0.8). A significance level of 5% was applied to all analyses. 

## 3. Results

### 3.1. Patient nº1

A 30-year-old male patient was admitted to the hospital after suffering a mid-shaft fracture of the tibia, resulting from an ankle sprain followed by a fall from his own height. Despite the traumatic incident, the patient remained conscious, alert, and oriented upon arrival at the hospital. 

He was classified as ASA II, indicating a moderate physical condition and relatively good tolerance to anesthetic and surgical procedures. The patient’s social habits included alcohol consumption. The patient presented with a closed fracture in the right tibia, classified as OTA/AO 42C2, as a result of the ankle sprain. No complications were reported during the surgery, and Dipyrone (500 mg every 6 h for 7 days), Tramadol (50 mg every 8 h for 7 days), Rivaroxaban (10 mg every 24 h for 20 days), and Cephalexin (500 mg every 6 h for 7 days) were prescribed for this patient.

Table 1 provides further information about the anthropometric and hematological parameters as well as X-ray images during the preoperative period. Figure 2 presents the results from ALP and RUST at the three time points for this patient. Both surgeons provided similar answers regarding RUST, which increased by 25% in T2 and 50% in T3 when compared to T1. ALP levels also increased by 75.5% and 85.7%, respectively, for T2 and T3 when compared to T1. Overall, for this patient, the RUST scores and ALP levels had the same behavior post tibial reamed intramedullary nailing surgery.

### 3.2. Patient nº2

Patient 2, a 54-year-old man, was admitted to the hospital after an accident involving his motorcycle and a car. During the incident, he sustained an open fracture in the diaphysis of the right tibia, classified as OTA/AO 42C2. 

Upon admission, the patient remained conscious, alert, and oriented, and was classified as ASA I, indicating a good physical condition. Pre-surgical blood tests are shown in Table 2. No complications were reported during the surgery and Dipyrone (500 mg every 6 h for 7 days), Tramadol (50 mg every 8 h for 7 days), Rivaroxaban (10 mg every 24 h for 20 days), Ciprofloxacin (500 mg every 12 h for 14 days), and Clindamycin (600 mg every 6 h for 14 days) were prescribed for this patient. 

Figure 3 presents the results from ALP and RUST for the three time points for this patient. Surgeons only disagreed in T3 regarding RUST, but the mean score from the two surgeons (5.5) at T3 was 37.5% higher than at the other time points. ALP levels increased by 142.2% at T2 when compared to T1. However, ALP at T3 decreased by 8.3% when compared to T2. For this patient, the association between the RUST score and ALP levels was only evident on the 49th day after tibial reamed intramedullary nailing surgery.

### 3.3. Patient nº3

Patient 3, a 35-year-old woman, was admitted to the hospital after sustaining an open fracture in the diaphysis of the right tibia, classified as OTA/AO 42C3, during a jiu-jitsu training session. 

At the time of admission, the patient remained conscious, alert, and oriented. Her physical status was assessed as ASA I, indicating generally good physical health. Blood test results are shown in Table 3. No complications were reported during the surgery and Dipyrone (500 mg every 6 h for 7 days), Tramadol (50 mg every 8 h for 7 days), Rivaroxaban (10 mg every 24 h for 20 days), Ciprofloxacin (500 mg every 12 h for 14 days), and Clindamycin (600 mg every 6 h for 14 days) were prescribed for this patient. 

Figure 4 presents the results for ALP and RUST at the three time points for this patient. Surgeons disagreed at some points for T2 and T3, but the overall scores indicate increases of 12.5% and 37.5%, respectively, for T2 and T3 when compared to T1. Similar to patient 2, after an increase of 76.5% at T2 when compared to T1 regarding ALP, ALP at T3 was decreased (9.6%) when compared to T2. For this patient, the concordance between RUST score and ALP levels occurred at both the 21st and 49th days after tibial reamed intramedullary nailing surgery.

### 3.4. Patient nº4

Patient 4, a 45-year-old woman, was admitted to the hospital following an accident involving her motorcycle and a car. During the incident, she sustained an open fracture in the diaphysis of the right tibia, classified as OTA/AO 42A3. 

Upon arrival at the hospital, the patient remained conscious, alert, and oriented. She was classified as ASA II, indicating a moderate physical condition. The patient reported alcohol consumption as part of her social habits. Blood test results indicated that the patient had significant anemia, indicated by low values of red blood cells, hemoglobin, and hematocrit (Table 4). The elevated white blood cell count suggested a possible inflammatory or infectious response, while the presence of band cells could have indicated a bacterial infection. No complications were reported during the surgery and Dipyrone (500 mg every 6 h for 7 days), Tramadol (50 mg every 8 h for 7 days), Rivaroxaban (10 mg every 24 h for 20 days), Ciprofloxacin (500 mg every 12 h for 14 days), and Clindamycin (600 mg every 6 h for 14 days) were prescribed for this patient. 

Figure 5 presents the results from ALP and RUST at the three time points for this patient. Surgeons only disagreed at T3 regarding RUST, which was similar at T1 and T2. At T3, the score increased by 12.5%. In contrast, ALP increased by 171.9% and 253.1%, respectively, at T2 and T3 when compared to T1. For this patient, the concordance between the RUST score and ALP levels occurred on the 49th day after tibial reamed intramedullary nailing surgery.

### 3.5. Variance Analysis

Data from the four patients were considered for the group comparison, which is shown in Figure 6. ANOVA for repeated measures indicated a significant effect for time points regarding ALP (*p* = 0.002; F = 20.61). The ALP levels were distinct from T1 when compared to T2 (*p* = 0.005) or T3 (*p* = 0.003), but no significant differences were observed between the latter two time points (*p* = 1.000). Moreover, high ES and percentage differences were observed between T1 and the other time points (T2: ES = 4.89; % = 116.6; T3: ES = 4.36; % = 130.2). Regarding RUST, due to the absence of variance in T1, it was not possible to conduct the repeated-measures ANOVA. However, increases of 9.3% and 34.3%, respectively, were observed for T2 and T3 when compared to T1.

## 4. Discussion

The results herein demonstrated that both RUST and ALP contribute to interpretations regarding bone consolidation after surgery for tibial diaphysis fracture repair. While our initial analysis of variance shows that ALP levels are elevated at 21 or 49 days compared to T1, it is essential to consider individual patient factors. In two cases, ALP levels began to decrease at T3, whereas they continued to rise in the remaining cases. However, when aligned with RUST scores, the scores increased at T3 in all cases, indicating concordance between these two sets of data.

Chaudhary et al. [27] monitored 44 patients with long bone fractures (including the tibia, femur, humerus, radius, and ulna) over a period of 24 weeks to assess serum ALP levels. The authors noted a transient elevation in ALP concentration, peaking within the first seven weeks post-intervention for all patients, followed by a subsequent normalization of levels in those exhibiting signs of bone consolidation. Based on this study, we considered monitoring ALP and RUST over 49 days. However, it is possible to observe different trends in ALP behavior between the 3rd and 7th weeks among the analyzed patients. While ALP starts to decline in the 7th week for patients 2 and 3, it continues to increase for patients 1 and 4. Factors such as trauma severity, initial patient condition, and postoperative therapeutic approaches could help to explain the different behaviors of ALP after surgery. Although patients 2 and 3 presented with transverse fractures of moderate and high severity, the same cannot be said for patients 1 and 4. The severity of the fracture for patient 4 was substantially higher than that of patient 1; however, ALP continued to increase in both cases. Regarding the initial condition, a similar discussion can be provided. In addition to the fracture, patient 4 presented a possibly associated bacterial infection, which was a distinct scenario to that of patient 1. Nevertheless, all patients received the same medication prescriptions after surgery, which may not have affected recovery in the postoperative period.

Although this result contrasts with that of Chaudhary et al. [27], it aligns with those of Oni et al. [28], who observed substantial variations in ALP levels up to the 20th week of recovery, criticizing the use of this molecule as a marker of bone consolidation. These authors observed that fractures typically healing uneventfully exhibited higher levels of osteocalcin activity compared to fractures experiencing delayed union, suggesting a reduced osteoblastic activity in fractures with slower healing rates. Such findings put forward the evaluation of osteocalcin activity post fracture as a valuable prognostic indicator. In contrast, total serum ALP showed no significant differences between the injury groups or within the healing groups. A different conclusion, however, was drawn by Bowles et al. [29]. After monitoring twenty patients who suffered isolated fractures of the tibia and/or fibula over 20 weeks, the authors observed that after a decrease by day 4, ALP levels reached a nadir at day 8 and then presented a tendency to increase until the 20th week. A major difference in this study was that the ALP measured was the bone isoenzyme ALP (BS-ALP).

While the discussion on which ALP form may offer better diagnostic, prognostic, or monitoring capabilities for long bone fractures is beyond the scope of our conclusions, Ajai et al. [22] provide valuable insights regarding the measurement of total serum ALP for this context. After monitoring ninety-five patients with simple, fresh (<7 days) traumatic diaphyseal fractures of the tibia and fibula, the authors indicated that serum ALP activity was significantly associated with the healing progress of fractures. It was observed that fractures with delayed union exhibited higher levels of ALP activity, suggesting increased osteoblastic activity compared to fractures which were healing normally. However, no significant differences were found in total serum ALP activity between injury groups or within healing groups. These results underscore the potential of serum ALP as a useful prognostic indicator in the monitoring of fracture healing, especially in cases of delayed union.

The primary contribution and innovation of this study are not solely associated with the behavior of ALP following tibial diaphyseal fractures but rather with the association of this molecule with RUST. It is important to emphasize that although RUST has a subjective nature, our assessments were conducted by two doctors with extensive experience in tibial diaphyseal surgeries, thereby reducing the bias of this score. Additionally, a positive outcome to highlight is that disagreements between RUST assignments were observed in some instances, which is expected and consistent with the subjective nature of the scale. In this scenario, the RUST score consistently supported the bone consolidation process on the 49th day. Two points need to be highlighted in this context. First, the observation of changes in RUST during this period strengthens our decision to limit the monitoring of ALP and RUST to the postoperative period (i.e., the 49th day). The second insight revolves around the bone consolidation process as a whole. The maximum score on the RUST scale is 12, indicating complete bone consolidation. The highest score obtained in our sample, regardless of the evaluating surgeon, was 6 (patient 1). This suggests that a longer timeframe is required for more comprehensive monitoring.

The perspectives mentioned earlier are consistent with the metabolic and physiological context in which ALP is involved. During bone consolidation, ALP activity tends to increase, reflecting intense bone formation activity [20]. ALP facilitates the deposition of calcium phosphate, an essential component of the bone matrix, thus promoting union, mineralization, and the strengthening of the bone callus [21]. However, there is still limited discussion about the magnitude of ALP elevation or reduction in conjunction with RUST for the complementary analysis of bone consolidation. In this context, our group contributes a hypothesis that, although requiring future study for acceptance or rejection, this study sheds light on the association between RUST and ALP after tibial diaphyseal surgeries. When correlating raw RUST and ALP values (data obtained from all time points), a positive and weak correlation was observed (r = 0.43; *p* = 0.163). However, considering that as bone consolidation progresses and effective union consequently occurs, the delta of ALP values between T3 and T1 may provide more refined information about this process. When correlating the deltas between T3 and T1 from RUST and ALP values, a high and inverse correlation (r = −0.80; *p* = 0.195) was obtained. However, this correlation does not reach significance, most likely due to the small sample size. To confirm this, Figure 7 illustrates the different scenarios presented by patients 1 and 4 in this analysis. For patient 1, who received a score of 6 on the RUST scale from both surgeons at T3 and consequently showed an improvement in the progression of bone consolidation, the ALP delta between T3 and T1 (42 U/L) was lower than that of patient 4 (81 U/L), who presented lower RUST scores than patient 1 at T3. Although we have no data to support any suggestion of a biological mechanism, we speculate that a reduction in ALP (especially in association with an increase in RUST) may in part suggest that the osteoblastic activity is no longer exacerbated (since bone consolidation is almost completed).

Our study is not without limitations. Indeed, due to its small sample size of four case studies, our analysis of variance requires more samples. Despite this limitation, we would like to clarify that our study involves additional challenges in terms of recruiting patients (who were operated upon using the technique of tibial reamed intramedullary nailing) and monitoring clinical outcomes. Additionally, the hypothesis mentioned regarding the delta of ALP and its correlation with RUST may be dependent on the timing of these analyses. Another point to mention is the monitoring time. Although our results support the notion that seven weeks is a period which is capable of providing interpretations of ALP and RUST in the postoperative period, considering a longer timeframe (e.g., 20 weeks) may provide further insights into the associations suggested here. It is also important to highlight that patient 4 presented with anemia upon hospital admission. Despite her ALP showing absolute values (32 U/L) which were lower than those of the other patients (patient 1 = 49 U/L; patient 2 = 45 U/L; patient 3 = 47 U/L), we cannot assume that the fracture caused the anemic state. On the other hand, the ALP levels of patient 4 showed values close to those of the other patients at T2, with a slight reduction at T3. Our results do not allow us to suggest that anemia did not affect ALP levels in the postoperative period, meaning that further studies are necessary to verify this outcome. Nonspecific ALP levels can also be influenced by functions in other tissues, such as the liver and kidneys [20]. Thus, considering tissue-specific ALP [30] in future studies in association with RUST is a valid possibility to refine the relationship between these data. However, the limitations of our study do not diminish the innovations provided in this article. The literature surrounding ALP in tibial diaphyseal fractures, as well as in other long bones, is not extensive. Additionally, we did not find studies that associated serum ALP levels with RUST, which is a positive aspect that strengthens our study. Despite all of the strengths and limitations provided by this study, we affirm that a pathway of possibilities is opened up for future studies to refine the interpretation of bone consolidation after tibial diaphyseal fractures.

## 5. Conclusions

Serum ALP levels increased by the third postoperative week following tibial reamed intramedullary nailing surgery. While this increase persisted into the seventh week for two patients, a decline was observed in the same period for the other two patients. Considering that delta ALP (T3-T1 value) was lower in patients who exhibited the highest RUST score, we suggest that a synchronized analysis of ALP and RUST will allow medics to diagnose bone consolidation. Therefore, it can be concluded that the analysis of ALP alongside RUST may be complementary for evaluating bone consolidation following tibial reamed intramedullary nailing surgery, but future studies are needed to confirm this assertion.

## Figures and Tables

**Figure 1 biology-13-00407-f001:**
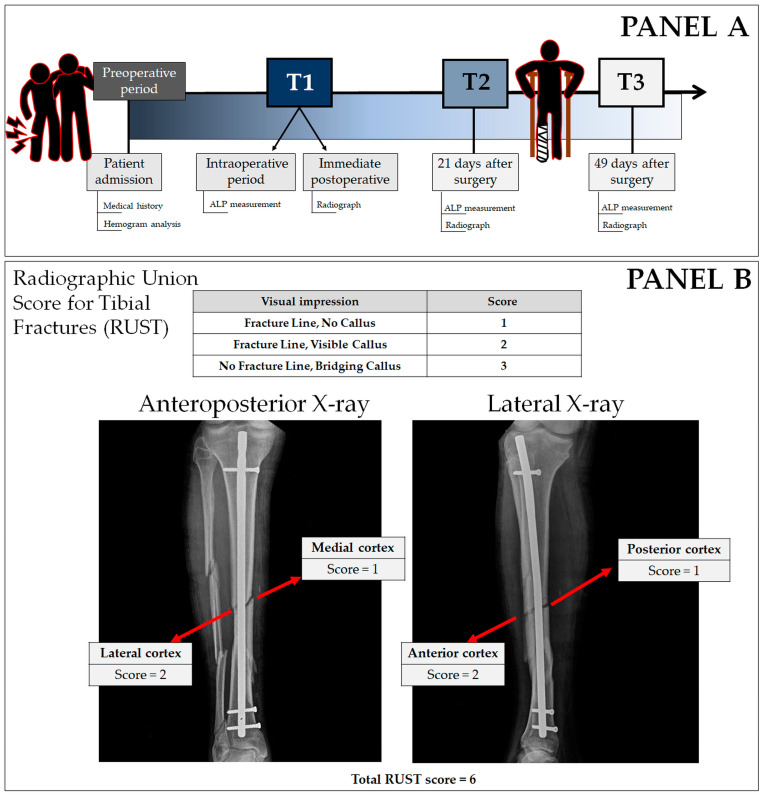
Panel (**A**) illustrates the experimental design. The RUST system, which assigns a score to anteroposterior (AP) and lateral (LAT) radiographs, is represented in panel (**B**).

**Figure 2 biology-13-00407-f002:**
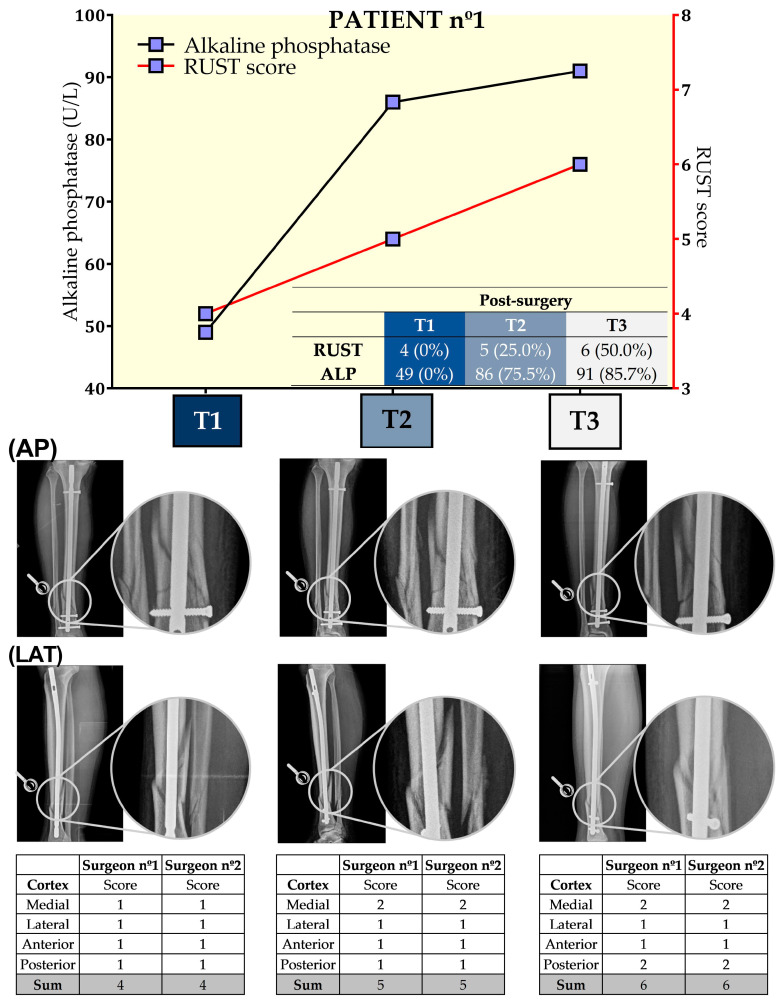
Seric alkaline phosphatase (left *y*-axis) and Radiographic Union Scale for Tibial Fractures (RUST) scores (right *y*-axis) from patient nº1 at T1, 21st (T2), and 49th (T3) days after tibial reamed intramedullary nailing surgery. AP and LAT refer to the anteroposterior and lateral radiographic images, respectively. The table within the graph provides the absolute values. In addition, data are relative (%) to T1, which was set at 0%.

**Figure 3 biology-13-00407-f003:**
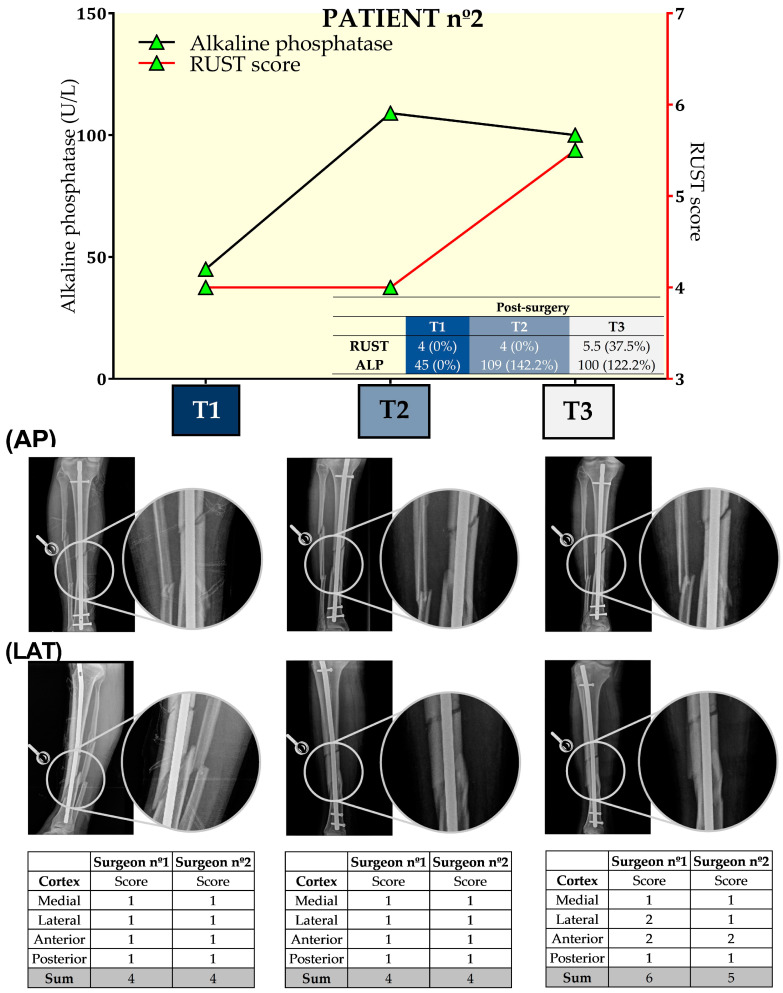
Seric alkaline phosphatase (left *y*-axis) and Radiographic Union Scale for Tibial Fractures (RUST) scores (right *y*-axis) from patient nº2 at T1, 21st (T2), and 49th (T3) days after tibial reamed intramedullary nailing surgery. AP and LAT refer to the anteroposterior and lateral radiographic images, respectively. The table within the graph provides the absolute values. In addition, data are relative (%) to T1, which was set at 0%.

**Figure 4 biology-13-00407-f004:**
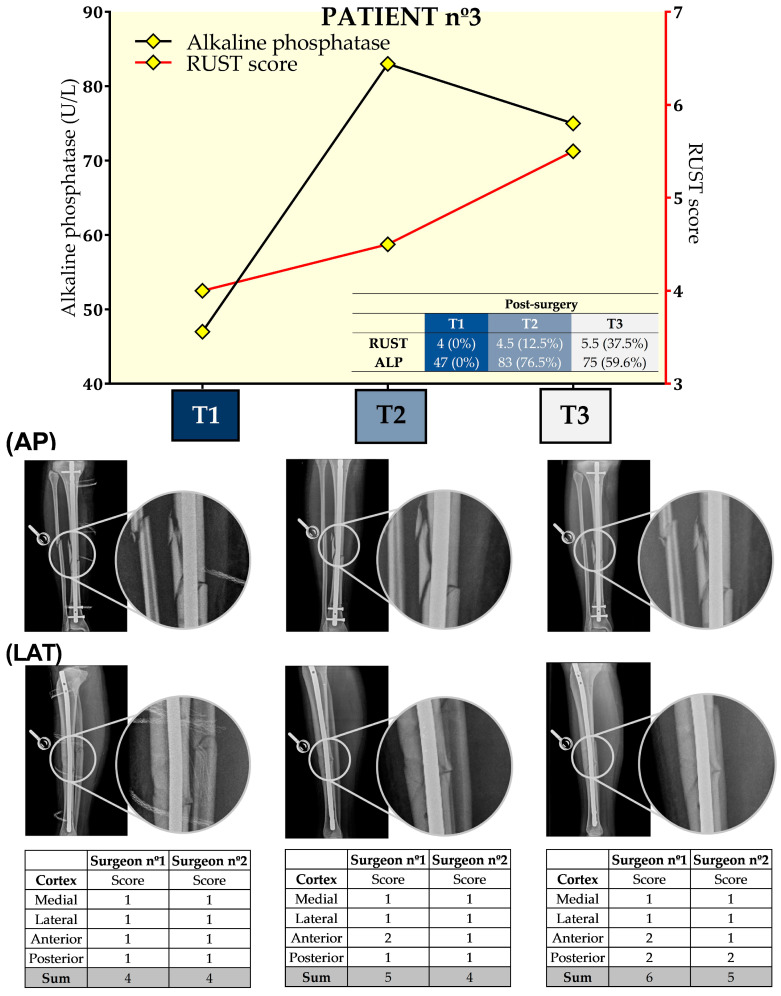
Seric alkaline phosphatase (left *y*-axis) and Radiographic Union Scale for Tibial Fractures (RUST) scores (right *y*-axis) from patient nº3 at T1, 21st (T2), and 49th (T3) days after tibial reamed intramedullary nailing surgery. AP and LAT refer to the anteroposterior and lateral radiographic images, respectively. The table within the graph provides the absolute values. In addition, data are relative (%) to T1, which was set at 0%.

**Figure 5 biology-13-00407-f005:**
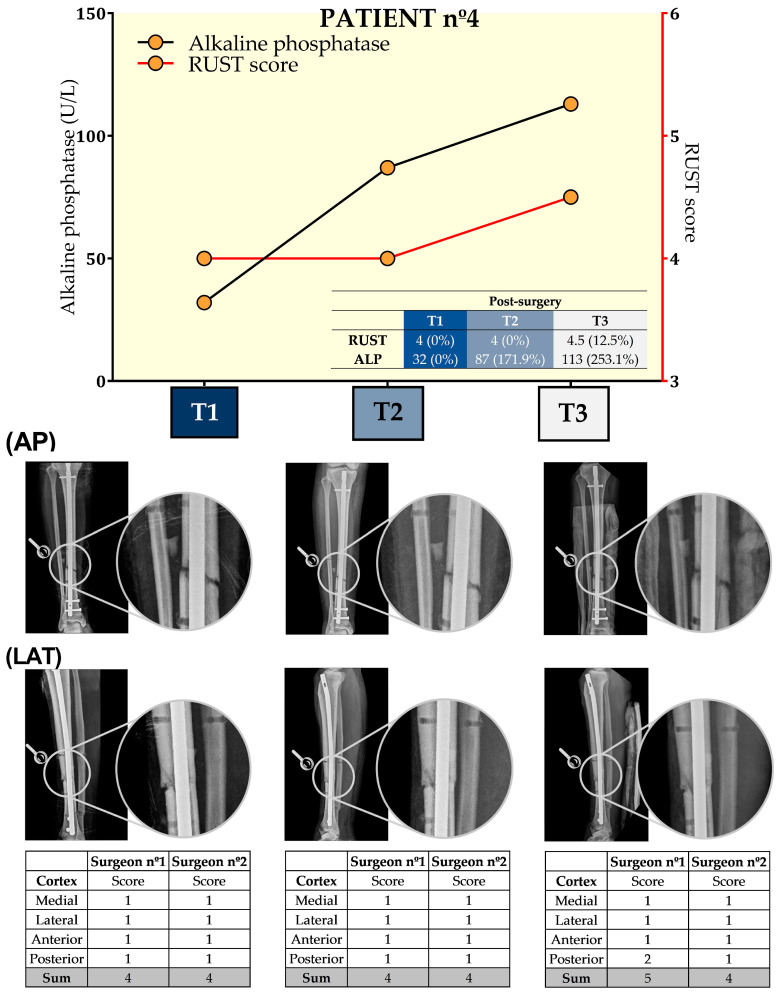
Seric alkaline phosphatase (left *y*-axis) and Radiographic Union Scale for Tibial Fractures (RUST) scores (right *y*-axis) from patient nº4 at T1, 21st (T2), and 49th (T3) days after tibial reamed intramedullary nailing surgery. AP and LAT refer to the anteroposterior and lateral radiographic images, respectively. The table within the graph provides the absolute values. In addition, data are relative (%) to T1, which was set at 0%.

**Figure 6 biology-13-00407-f006:**
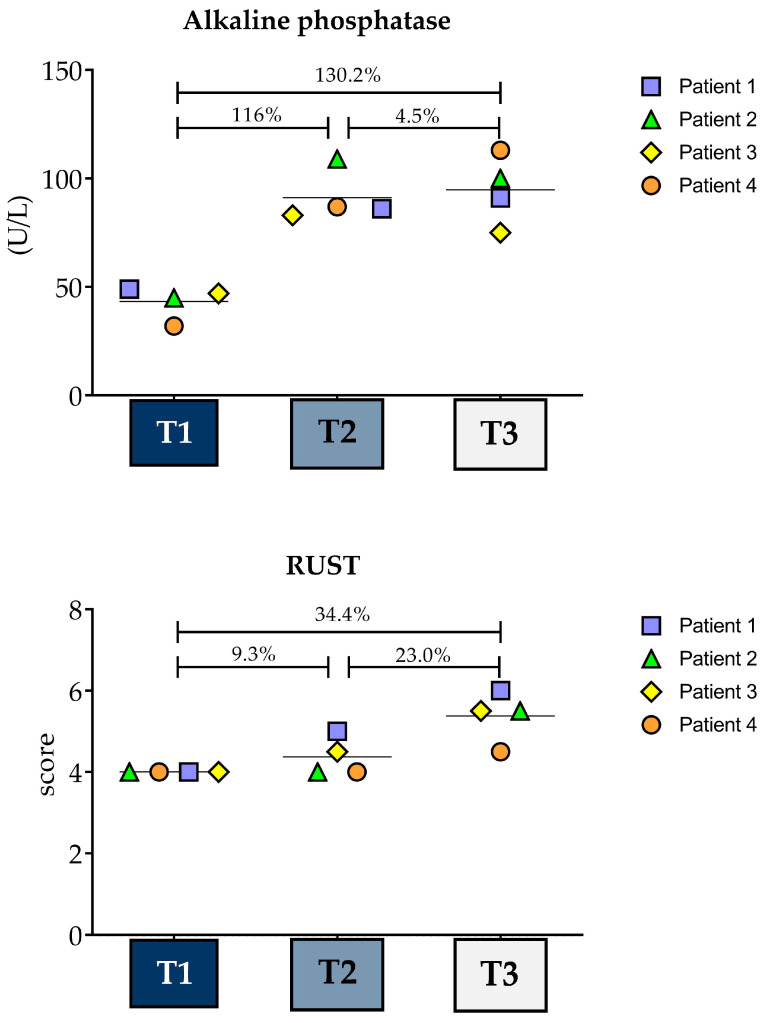
Comparisons considering data from the four patients. Each patient is represented by a different symbol and the horizontal lines represent the mean at T1, 21st (T2), and 49th (T3) days after tibial reamed intramedullary nailing surgery.

**Figure 7 biology-13-00407-f007:**
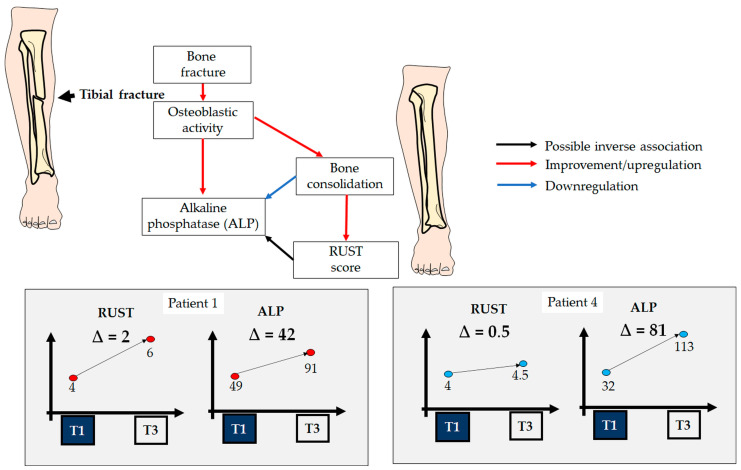
A didactical illustration showing possible links among the studied variables. Patients showing the highest ∆ RUST and the lowest ∆ ALP may suggest better bone consolidation, as is the case of patient nº1 in comparison with patient nº4.

**Table 1 biology-13-00407-t001:** Anthropometric and hematological data as well as X-ray images from patient nº1 at the preoperative period.

**Parameter**	**Result**	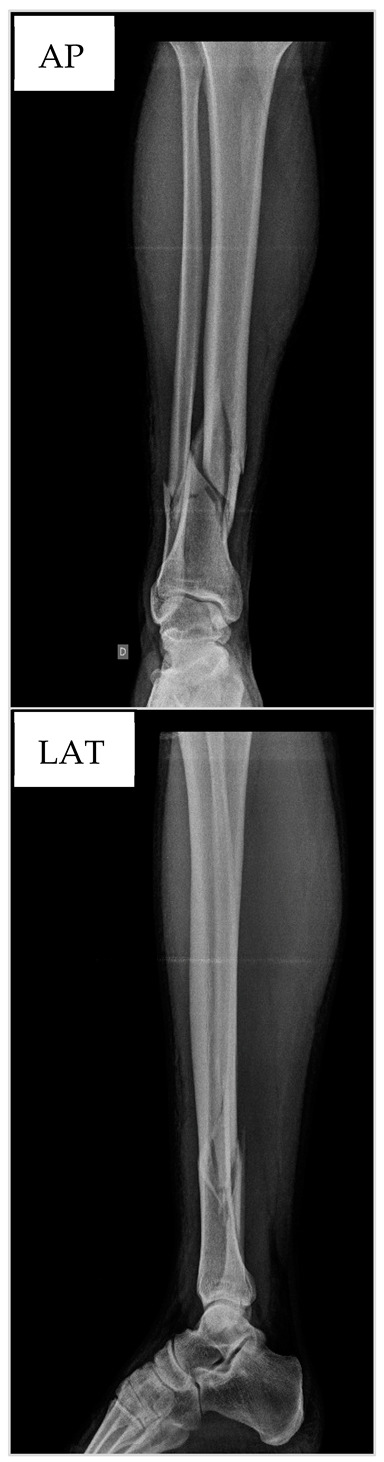
Body mass (kg)	82	
Height (m)	170	
BMI (kg/m^2^)	28.3	
RBC (10^6^/µL)	4.6	
Hemoglobin (g/dL)	13.2	
Hematocrit (%)	39.2	
MCV (fL)	85.2	
MCH (pg)	28.7	
MCHC (10^6^/µL)	33.7	
RDW (%)	13.6	
WBC (10^9^/µL)	8400	
Promyelocytes (%)	0	
Myelocytes (%)	0	
Metamyelocytes (%)	0	
Band cells (%)	0	
Segregated cells (%)	72	
Neutrophils (%)	72	
Eosinophils (%)	1	
Basophils (%)	0	
Typical lymphocytes (%)	18	
Atypical lymphocytes (%)	0	
Monocytes (%)	9	
Blasts (%)	0	
Platelets (10^9^/L)	211.000	
MPV (fL)	8.7	

**RBC**—red blood cells; **MCV**—mean corpuscular volume; **MCH**—mean corpuscular hemoglobin; **MCHC**—mean corpuscular hemoglobin concentration; **RDW**—red cell distribution width; **WBC**—white blood cells; **MPV**—mean platelet volume; **BMI**—body mass index; **AP**—anteroposterior; **LAT**—lateral.

**Table 2 biology-13-00407-t002:** Anthropometric and hematological data as well as X-ray images from patient nº2 at the preoperative period.

**Parameter**	**Result**	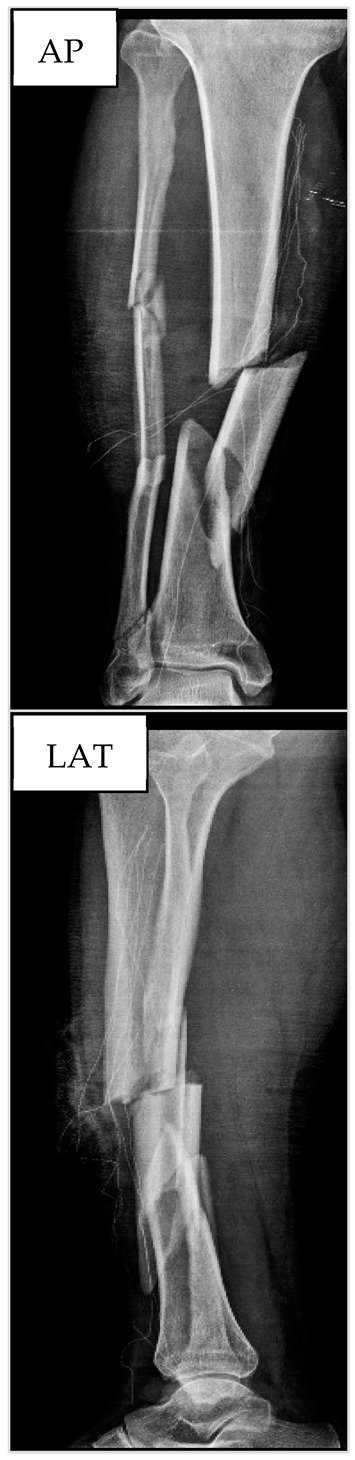
Body mass (kg)	82.0	
Height (m)	1.75	
BMI (kg/m^2^)	26.7	
RBC (10^6^/µL)	3.93	
Hemoglobin (g/dL)	12.5	
Hematocrit (%)	37.1	
MCV (fL)	94.4	
MCH (pg)	31.8	
MCHC (10^6^/µL)	33.7	
RDW (%)	11.5	
WBC (10^9^/µL)	9.700	
Promyelocytes (%)	0	
Myelocytes (%)	0	
Metamyelocytes (%)	0	
Band cells (%)	0	
Segregated cells (%)	73	
Neutrophils (%)	73	
Eosinophils (%)	1	
Basophils (%)	0	
Typical lymphocytes (%)	21	
Atypical lymphocytes (%)	0	
Monocytes (%)	5	
Blasts (%)	0	
Platelets (10^9^/L)	165.000	
MPV (fL)	10.5	

**RBC**—red blood cells; **MCV**—mean corpuscular volume; **MCH**—mean corpuscular hemoglobin; **MCHC**—mean corpuscular hemoglobin concentration; **RDW**—red cell distribution width; **WBC**—white blood cells; **MPV**—mean platelet volume; **BMI**—body mass index; **AP**—anteroposterior; **LAT**—lateral.

**Table 3 biology-13-00407-t003:** Anthropometric and hematological data as well as X-ray images from patient nº3 at the preoperative period.

**Parameter**	**Result**	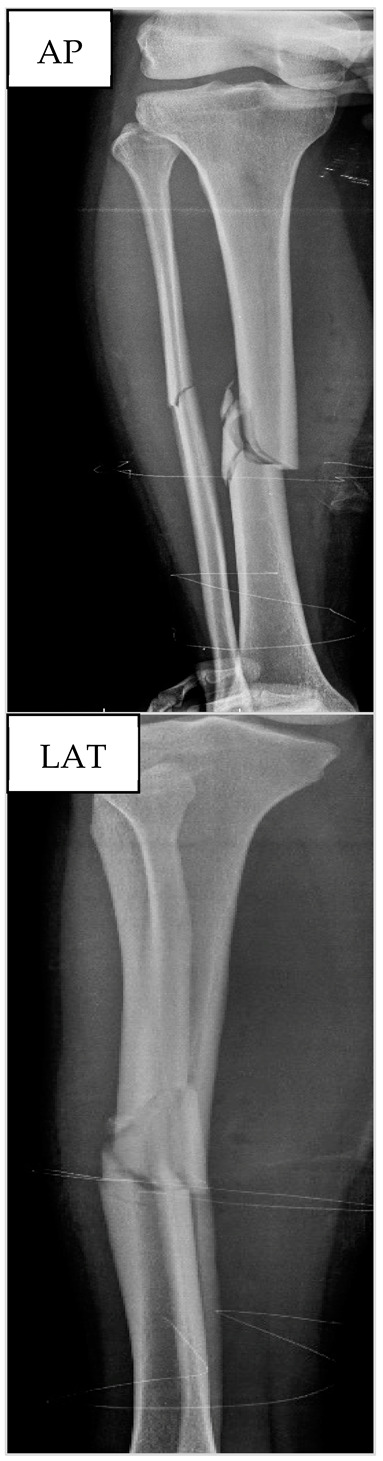
Body mass (kg)	63.0	
Height (m)	1.61	
BMI (kg/m^2^)	24.3	
RBC (10^6^/µL)	4.37	
Hemoglobin (g/dL)	12.8	
Hematocrit (%)	37.1	
MCV (fL)	84.9	
MCH (pg)	29.3	
MCHC (10^6^/µL)	34.5	
RDW (%)	13	
WBC (10^9^/µL)	9.400	
Promyelocytes (%)	0	
Myelocytes (%)	0	
Metamyelocytes (%)	0	
Band cells (%)	0	
Segregated cells (%)	71	
Neutrophils (%)	71	
Eosinophils (%)	1	
Basophils (%)	0	
Typical lymphocytes (%)	22	
Atypical lymphocytes (%)	0	
Monocytes (%)	6	
Blasts (%)	0	
Platelets (10^9^/L)	265.000	
MPV (fL)	9.4	

**RBC**—red blood cells; **MCV**—mean corpuscular volume; **MCH**—mean corpuscular hemoglobin; **MCHC**—mean corpuscular hemoglobin concentration; **RDW**—red cell distribution width; **WBC**—white blood cells; **MPV**—mean platelet volume; **BMI**—body mass index; **AP**—anteroposterior; **LAT**—lateral.

**Table 4 biology-13-00407-t004:** Anthropometric and hematological data as well as X-ray images from patient nº4 at the preoperative period.

**Parameter**	**Result**	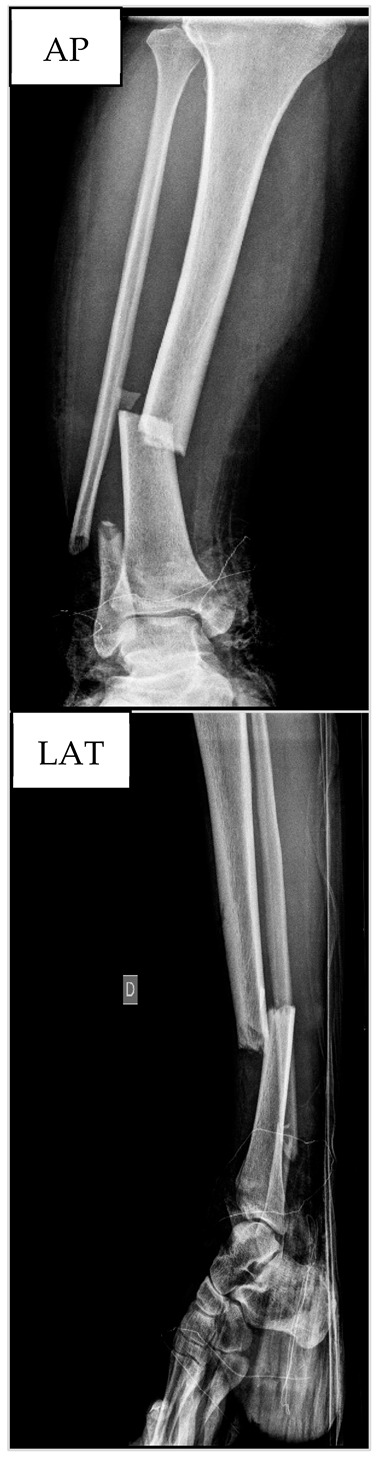
Body mass (kg)	74.0	
Height (m)	1.68	
BMI (kg/m^2^)	26.2	
RBC (10^6^/µL)	2.37	
Hemoglobin (g/dL)	7.2	
Hematocrit (%)	22.1	
MCV (fL)	93.2	
MCH (pg)	30.4	
MCHC (10^6^/µL)	32.6	
RDW (%)	12.7	
WBC (10^9^/µL)	15.100	
Promyelocytes (%)	0	
Myelocytes (%)	0	
Metamyelocytes (%)	0	
Band cells (%)	3	
Segregated cells (%)	69	
Neutrophils (%)	72	
Eosinophils (%)	0	
Basophils (%)	0	
Typical lymphocytes (%)	21	
Atypical lymphocytes (%)	0	
Monocytes (%)	7	
Blasts (%)	0	
Platelets (10^9^/L)	446.000	
MPV (fL)	7.9	

**RBC**—red blood cells; **MCV**—mean corpuscular volume; **MCH**—mean corpuscular hemoglobin; **MCHC**—mean corpuscular hemoglobin concentration; **RDW**—red cell distribution width; **WBC**—white blood cells; **MPV**—mean platelet volume; **BMI**—body mass index; **AP**—anteroposterior; **LAT**—lateral.

## Data Availability

The data are not stored in public repositories, but they can be provided upon request.

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
