# Peer review of "Comparison of Radiographic Outcomes Assessed via the Radiographic Union Scale for Tibial Fractures and Alkaline Phosphatase Levels during the Tibial Healing Process: A Series of Case Reports"

_biology, 2024, doi:10.3390/biology13060407_

Round 1

Reviewer 1 Report

Comments and Suggestions for Authors

The manuscript gives an interesting preliminary insight into the potential use of ALP along with RUST score in the assessment of tibial fracture healing in patients who underwent intramedullary nailing. I have a number of comments.

  1. The description of the particular cases can be shortened. There is no need to present a list of signs and symptoms or elements of anamnesis which were negative.

  2. The design of the study should be given as „case series”.

  3. Longer lasting studies involving more subjects are needed to determine whether ALP levels can be used in the diagnosis of delayed union or nonunion.

  4. Line 29: „with tibial fractures requiring surgery” instead of „surgical tibial fractures” should be written.

  5. Line 32: „ap-appointments” is written. Please, correct.

  6. Line 38, line 583: „medics” should be written instead of „medicals”.

  7. Line 52: Please, give the time frame for the diagnosis of a nonunion.

  8. Line 54: What is the incidence of nonunion in tibial fractures?

  9. Line 62: „However” should be written instead of „conversely”.

  10. Line 72: „Patients who sustained fractures” would sound better than „fractured patients”.

  11. Line 80: „Tibial fracture recovery” should be written.

  12. Preoperative ALP level assessment (T0) can not be used interchangeably with T1. When was the T0 examination performed – just before the surgery as it is written in the manuscript or 2 days prior (at admission) as can be concluded from Figure 1? Figure 2, 3, 4 captions: It can not be stated that there was an „immediate postoperative assessment” of ALP since the blood was taken before surgery. Please, change to „perioperative assessment”. The same applies to the statement in line 449.

  13. Line 176: „not completely healed” means that the fracture is partially healed. Score 4 indicates that „the fracture is completely not healed”. Please, change.

  14. The manuscript requires analysis by a statistical editor.

  15. „Preoperatory moment” should be changed to „preoperative period” throughout the manuscript.

  16. Was it determined what caused anaemia and infection in patient 4? Was it due to the sustained injury? Is it possible that those factories could have had an impact on ALP levels?

  17. Line 463: The meaning of the sentence beginning with „However” in unclear.

  18. Line 475: „healing uneventfully” should be written instead of „healing together”.

  19. Line 488: „ninety” should be written

  20. Line 519: „this study sheds light” should be written.

Comments on the Quality of English Language

Minor editing of English language required.

Reviewer 2 Report

Comments and Suggestions for Authors

In this manuscript, the authors provided a case study of four patients who underwent tibial reamed intramedullary nailing surgery to the show the correlation of RUST (Radiographic Union Scale for Tibial Fractures) scores and ALP (alkaline phosphatase) levels in the blood, to assess the use of blood ALP concentration as a postoperative monitoring tool. To make the manuscript contents clearer to the readers, I have the following questions and comments for the consideration of authors:

1.       To my knowledge, bone tissues are not the only source of ALP in the human body. How did the authors rule out the ALP concentration change due to the ALP from other sources, e.g the liver? Would this be a reason for the weak correlation between RUST and ALP values?

2.       As reported in Section 2.4, the radiograph analysis was performed by two experienced surgeons. The RUST scores agreed well for patient 1 but not others especially at the last time point (T3).  Would it be beneficial to have the radiographs scored by a radiologist?

3.       Please check the meaning of ‘nineth five’ in line 488.

Comments on the Quality of English Language

Careful proofreading and editing would help improve the readability and professional quality of the writing.
